# The Effect of Age on Non-Invasive Hemodynamics in Chronic Heart Failure Patients on Left-Ventricular Assist Device Support: A Pilot Study

**DOI:** 10.3390/jcm12010029

**Published:** 2022-12-20

**Authors:** Else-Marie van de Vreede, Floor van den Berg, Parsa Jahangiri, Kadir Caliskan, Francesco Mattace-Raso

**Affiliations:** 1Department of Geriatric Medicine, Erasmus MC University Medical Center, 3015 GD Rotterdam, The Netherlands; 2Department of Cardiology, Erasmus MC University Medical Center, 3015 GD Rotterdam, The Netherlands

**Keywords:** aging, non-invasive hemodynamics, blood pressure, heart failure, continuous flow, LVAD, non-pulsatility, adverse events

## Abstract

Background: Implantation of continuous flow left ventricular assist devices (LVAD’s) has been increasingly used in patients with advanced heart failure (HF). Little is known about the non-invasive hemodynamics and the relationship with adverse events in this specific group of patients. We aimed to identify any differences in non-invasive hemodynamics in patients with an LVAD in different age categories and to investigate if there is an association with major adverse events. Methods: In this observational cross-sectional study, HF patients with a continuous flow LVAD were included. Non-invasive hemodynamic parameters were measured with a validated, automated oscillometric blood pressure monitor. The occurrences of adverse events were registered by reviewing the medical records of the patients. An independent-samples T-test and Chi-square test were used to compare different groups of patients. Results: Forty-seven patients were included; of these, only 12 (25.6%) had a successful measurement. Heart rate, heart rate-adjusted augmentation index, and pulse wave velocity were higher in the ≥55 years of age LVAD group compared to the <55 years of age LVAD group (all *p* < 0.05). Stroke volume was significantly lower in the ≥55 years of age LVAD group compared to the <55 years of age LVAD group (*p* = 0.015). Patients with adverse events such as cardiovascular events, GI-bleeding, or admission to a hospital had lower central pulse pressure (cPP) than patients without any adverse event. Conclusion: Older LVAD patients have a significantly higher heart rate, heart rate-adjusted augmentation index, and pulse wave velocity and a significantly lower stroke volume compared to participants aged < 55 years. The pulsatile component of blood pressure was decreased in patients with adverse events.

## 1. Introduction

Implantation of a left-ventricular assist device (LVAD) has been increasingly used in advanced heart failure (HF) patients that are refractory to optimal medical therapy. It can be used as a bridge-to-heart transplantation therapy, or as destination therapy [1]. Bridge-to-heart transplantation therapy reduces mortality and improves patients’ overall condition [2]. Patients undergoing destination therapy are, by definition, ineligible for transplantation, mostly due to increased age or other comorbidities [3]. Heart failure remains a potentially fatal disease despite advances in therapy [4]. If evolved therapies for advanced heart failure patients give no result, LVAD implantation can be used. The small pool of donor hearts and the growing group of heart failure patients in need of a transplant causes an increase in LVAD destination therapy [1]. 

LVADs improves clinical outcomes in HF patients by improving hemodynamics, but with a non-physiological non-pulsatile, continuous-flow hemodynamics [5]. Little is known about the hemodynamics patterns in patients implanted with an LVAD. Because of the improvement of continuous-flow LVADs (CF-LVADs) in durability, the cumulative effects of these changed hemodynamics have become an important area of interest [6]. Despite improvements in survival, CF-LVADs are associated with complications such as long-term end-organ dysfunction, gastro-intestinal (GI) bleeding, hospitalization, and cardiovascular events. Both low arterial pulsatility and increased aortic stiffness may increase the risk of those adverse events [7,8,9]. 

Clarification of hemodynamic parameters in LVAD patients may help to optimize these parameters and improve clinical outcomes. While there is no conclusive evidence to suggest that this will occur, we hypothesize that LVAD patients will have a specific blood pressure profile that differs from patients without an LVAD. Therefore, we will investigate non-invasive hemodynamic parameters in patients with an LVAD. The purpose of this study is to identify any differences in non-invasive hemodynamics between patients with an LVAD within different age categories and whether specific hemodynamic patterns are associated with any adverse event. 

## 2. Materials and Methods

### 2.1. Patient Selection

We included outpatients with HF with an LVAD who were followed by the department of Cardiology at the Erasmus MC University Medical Center, Rotterdam, in the period of December 2021 until March 2022. Patients were asked to participate in the study if they were >18 years when they were implanted with an LVAD. Written informed consent was obtained from all participants before we started the measurements. The study protocol (MEC-2015-405) was approved by the medical-ethical board of Erasmus MC University Medical Center, Rotterdam. 

### 2.2. Variables

Demographic data used: age, sex, weight, length, BMI (kg/m^2^), etiology of HF, indication for LVAD, and device type; medications were assessed by reviewing the medical records of the patients. Renal function (assessed by eGFR–creatinine based), bilirubin, NT-proBNP, and hemoglobin were measured during laboratory control.

### 2.3. Hemodynamic Data

LVAD device values and blood pressure values were collected by reviewing the electronic medical records of the patients. Non-invasive hemodynamic data were collected through measurements with a Mobil-O-Graph. The Mobil-O-Graph (IEM, Rheinland, Germany) is a previously validated, automated oscillometric blood pressure monitor [10,11]. Besides brachial blood pressure readings, the monitor provides an estimation of the aortic pulse wave velocity (aPWV) through analysis of brachial pulse wave morphology. The measurements were performed in a quiet examination room, with the patient seated with both legs uncrossed on the floor, after two minutes rest. The brachial cuff was applied to the right upper arm on the right spot of the brachial artery and inflated two times to perform a complete measurement. 

Other hemodynamic parameters obtained during the same measurement, using inbuilt algorithms, included mean arterial pressure (MAP), heart rate (HR), pulse pressure (PP), stroke volume (SV), cardiac output (CO), peripheral vascular resistance (SVR), cardiac index (CIx), augmentation pressure, and augmentation index (AIx). The collected data were uploaded into HMS Client Server 5.1 software, Rheinland, Germany. 

### 2.4. Adverse Events Registration

We retrospectively reviewed the electronic medical records of patients who were selected in the study to register if there were adverse events in people on the waiting list or after LVAD implantation. We focused on three major complications. Adverse events were defined as cardiovascular events such as myocardial infarct or CVA, gastro-intestinal bleeding, and admission to a hospital.

### 2.5. Statistical Analysis

Statistical analyses were performed using IBM SPSS Statistics 28, Chicago, IL, USA, version for Windows. All patients with a successful or failed measurement were analyzed for baseline characteristics and LVAD device values. For further analyses, patients without a successful measurement were excluded. Normally distributed data are presented as median and interquartile range (IQR) for continuous variables and frequency (%) for categorical variables. Characteristics were compared between the group patients with and without complications, using the T-test for continuous variables. The Chi-square test was used to compare percentages. To investigate the possible role of aging patients, patients were divided into age categories < 55 years and ≥55 years. A two-tailed *p*-value of ≤0.05 was considered statistically significant. 

## 3. Results

Figure 1 shows the flowchart of the inclusion of patients. In total, 47 patients were asked to participate in the study. All 47 patients underwent a blood pressure measurement with the Mobil-O-Graph. Thirty-five patients were excluded from analyses of the non-invasive hemodynamics because we were not able to perform a successful measurement in these patients. We included 12 patients for analyses of the non-invasive hemodynamics. 

Table 1 shows the baseline characteristics for patients with HF with an LVAD with a successful blood pressure measurement, compared with patients with a non-successful measurement. In 25.6%, we were able to perform a successful measurement in HF with an LVAD. There was no statistically significant difference in age, sex, weight, BMI, time since LVAD implantation, or etiology of HF between groups. In both groups, most of the patients were men. Only one patient had an LVAD device type HM-II; all other patients had a device type HM-III. There was also no statistically significant difference between the blood pressure values measured by the LVAD nurse practitioner. The LVAD parameters flow and PI were significant different between groups. In patients with a failed measurement, we found a higher flow (*p* = 0.014) and a lower PI (*p* = 0.002). The mean NT-proBNP in patients with a failed measurement was higher; however, this difference was barely statistically significant (*p* = 0.057). 

Table 2 shows the baseline characteristics of patients with an LVAD in the different age categories, <55 years and ≥55 years. The <55 years group included four patients, the group with patients ≥ 55 years included seven patients. 

Figure 2 shows the mean levels and distribution of the non-invasive hemodynamic parameters in the age categories < 55 years and ≥55 years. Heart rate, heart rate-adjusted augmentation index, and pulse wave velocity were statistically significant higher in the ≥55 years of age LVAD group compared to the <55 years of age LVAD group (all *p* < 0.05). Stroke volume was significantly lower in the ≥55 years of age LVAD group compared to the <55 years of age LVAD group (*p* = 0.015). 

In Table 3, the baseline characteristics of patients with HF with an LVAD are presented. We included 12 patients with HF with an LVAD, of which seven patients had no adverse event and five patients had adverse event. Adverse events include hospitalization (for cardiac causes), GI-bleeding, and a cardiovascular event. There was no statistically significant difference in any of the baseline characteristics between the two groups. Patients with any adverse event did have a longer median time since LVAD implantation (13.5 versus 28.3 months). Only one patient had an LVAD device type HM-II, all other patients had an LVAD device type HM-III. The LVAD parameters, such as RPM, flow, PI, and pulse power, were measured by the LVAD technician on the same day our measurement was performed. This also applies for the blood pressure values and the lab results. There were no missing data for any of the patients.

In Figure 3, the mean values and the distribution of the non-invasive hemodynamic parameters are shown in different groups, with and without complications. Only the central pulse pressure (cPP) was statistically significant different between the two groups. Patients without adverse events had a median cPP of 18.5 IQR 14.3–24.3, patients with adverse events had a median cPP of 14 IQR 9–16, *p* = 0.044. There was no statistically significant difference between the mean values of the other hemodynamics. 

## 4. Discussion

In this cross-sectional study, we found that older LVAD patients had higher aortic stiffness, augmentation index, and heart rate but lower stroke volume compared to participants aged < 55 years. The pulsatile component of blood pressure was decreased in patients with previous adverse events. 

LVAD patients have a reduced arterial pulse pressure [12], which makes it difficult to measure the blood pressure with an automated oscillometric blood pressure monitor. Nonetheless, previous studies validated blood pressure measurements with the Mobil-O-Graph in LVAD patients using an A-line comparison. Even though this methodology was previously validated [10,13] with a success rate of 91% [13] and 82% [10], we were able to obtain the measurements in only 25.6% of the LVAD patients.

Castagna et al. found that a lower HeartMate II speed was associated with a higher success rate (*p* < 0.05) [10]. In our study, we also compared the RPM speed between the successful and failed measurement group. We found no significant difference, but our study group consisted of only one HeartMate II against 46 HeartMate III implants, which could make this comparison questionable. Furthermore, they found that no other measured parameters were associated with measurement success. Meanwhile, we did find that LVAD flow, LVAD Pulse Index (PI), and NT-proBNP were associated with measurement success. Lower values of the PI indicate lower filling pressures and lower contractility of the left ventricle. With lower values of the PI, the LVAD is providing greater support and the flow pulse coming from the left ventricle is lower [14]. This is a possible explanation why it is harder to measure the blood pressure in patients with a lower PI. 

A previous study by Schofield et al. [15] investigated a limited number of hemodynamic parameters in LVAD patients and is the only study who investigated this subject. Because of the limited existing literature about hemodynamics in LVAD patients, we compared our study findings to the literature regarding the association between hemodynamics and age in healthy individuals. First of all, Houghton et al. [16] found that stroke volume under resting conditions in healthy individuals reduces with increasing age. Our study also found a significantly lower stroke volume in LVAD patients ≥ 55 years compared to LVAD patients < 55 years. The same study of Houghton et al. [16] also investigated the association between aging and heart rate and found that there was no significant difference between heart rate in younger and older participants. Surprisingly, in our LVAD patients we did find a significant increase in heart rate with increasing age. PWV is a marker for arterial stiffness, which is an indicator for vascular aging. It can predict cardiovascular events according to several studies [17,18,19]. According to Styczynski et al. [20], there is an age-related increase in PWV. In our study we also found that LVAD patients ≥ 55 years had a significantly higher PWV than LVAD patients < 55 years. Thus, an increased age is associated with a higher PWV, which could function as a predictor for cardiovascular events. Notable is that the augmentation index is not significantly different in LVAD patients divided in different age categories in our study. Augmentation index is an indirect measure of arterial stiffness and is calculated with the augmentation pressure and pulse pressure. According to Fantin et al. [21], the AIx increases with age up to 55 years and tend to plateau thereafter. Because our age categories are divided into < 55 years and ≥ 55 years, this could explain why there is no significant difference due to the plateauing of the AIx increase. An associated hemodynamic parameter is the heart rate-adjusted augmentation index. To confound for the influence of heart rate, the augmentation index is calculated as it would be at 75 beats per minute. Beckmann et al. [22] found that the AIx@75 significantly increases with age. In our study, the AIx@75 also significantly increased between age categories <55 years and ≥55 years.

All non-invasive hemodynamic parameters were not significantly different between men and women. This outcome is difficult to judge because our study population only consisted of two women and nine men. Because of the small study group and imbalance, it is injudicious to conclude that there is truly no significant difference between men and women for all non-invasive hemodynamic parameters. 

The non-invasive hemodynamic parameters measured in the previous studies discussed above were investigated in participants without an LVAD and should therefore be taken into consideration. What is interesting to see is that some non-invasive hemodynamic parameters show the same difference in age categories in patients with and without an LVAD. This may highlight that, on some levels, the blood pressure profile of heart failure patients with and without an LVAD is very similar. However, there are also some differences between our study and the existing literature, such as heart rate, which indicates that LVAD patients do have a different blood pressure profile. 

As a result of a small pool of heart donors and the large growing group of heart failure patients in need of a transplant, destination therapy with an LVAD, which mainly consist of older patients > 60 years, is growing and accounts for progressively more implants than before [23]. Technological improvements in pump design and surgical techniques have made the efficiency of LVAD support longer [24]. As a result of the fast development in LVAD technology and the longer efficiency of LVAD support, understanding hemodynamics in LVAD patients has become more clinically relevant. In patients with HF with an LVAD, we found that a decrease in cPP in patients was associated with adverse events. This is consistent with findings in earlier studies that showed that adverse events in patients with an LVAD may be partly a result of chronic exposure to a low pulsatility. Patients with CF-LVADs with a low PP can have elevated levels of sympathetic nerve activity (SNA) [25], which, by causing high BP levels, predisposes to adverse events [26]. Furthermore, it has been suggested that higher levels of increased sympathetic tone can cause smooth muscle relaxation. This can lead to arteriovenous dilatation and ultimately arteriovenous malformation, which causes GI-bleeding [27]. Crow et al. retrospectively compared patients with pulsatile versus non-pulsatile LVADs and they found higher rates of GI-bleeding in patients with a non-pulsatile flow [28]. CF-LVADs give a continuous blood flow with minimal pulsatility that decreases the cyclical stretch of the arterial wall, leading to increased aortic stiffness [29]. It is reported that the aortic stiffness increases immediately after LVAD implantation, with attenuation of this increase in the first year [29]. Higher aortic stiffness is associated with significantly higher rates of common complications of CF-LVAD therapy [8].

Limitations: The findings of our study must be interpreted cautiously because of the relatively small group of patients included. While measuring, we were only able to successfully measure in a small percentage of our patients. The difficulty of measuring LVAD patients is most likely due to different flow. LVAD patients have a constant flow, while patients without an LVAD have a pulsatile flow. Application of non-invasive haemodynamic assessment based on an oscillometric blood pressure monitor might not be feasible in LVAD patients. This means that the LVAD represents an obstacle to carrying out valuable pulse wave analysis. We assessed patients at different times after LVAD implantation of since they were diagnosed with HF. Especially in patients with an LVAD, it is possible that in the early phase after LVAD implantation they were not completely stabilized and recovered from the device implantation. Because of the cross-sectional design, we did not perform serial assessments; therefore, we cannot rule out the possibility that the non-invasive hemodynamics may change over time. One strength of our explorative study is the examination of a relatively novel and increasing population with an increasing relevance in the future. Second, measurements with the Mobil-O-Graph only take a few minutes and could easily be applied in daily practice. Before this can be applied, research is necessary regarding the clinical relevance of the measured non-invasive hemodynamics.

## 5. Conclusions

In this cross-sectional study, we found that LVAD patients aged 55 years and older had higher levels of aortic stiffness, augmentation index, and heartrate and a lower stroke volume when compared with younger patients. The pulsatile component of blood pressure is decreased in patients with adverse events. 

Considering the explorative character of this study, future research is necessary to investigate whether the non-invasive hemodynamics can predict the occurrence of adverse events in patients with HF with an LVAD.

## Figures and Tables

**Figure 1 jcm-12-00029-f001:**
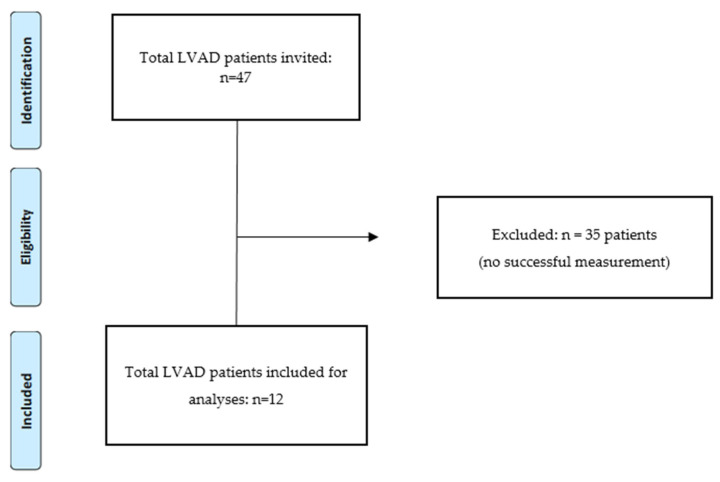
Flowchart inclusion of patients.

**Figure 2 jcm-12-00029-f002:**
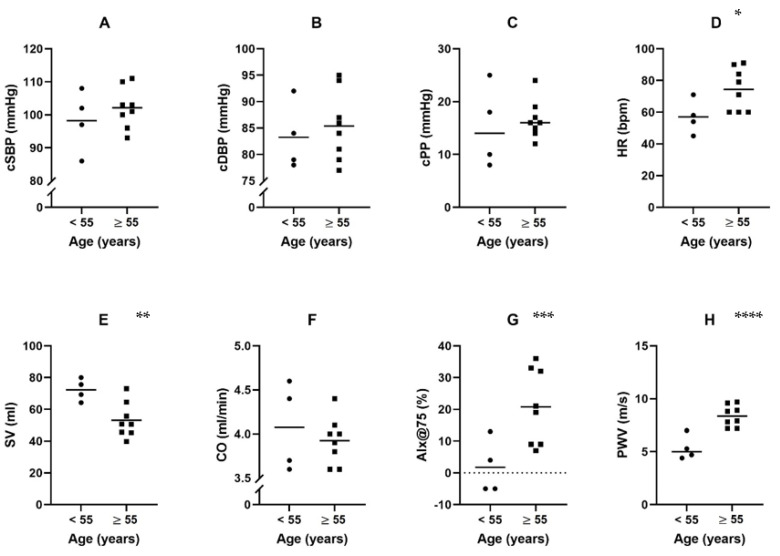
Distribution of non-invasive hemodynamics in HF with an LVAD within different age categories (**A**–**H**). Abbreviations: cSBP, central systolic blood pressure; cDBP, central diastolic blood pressure; cPP, central pulse pressure; HR, heart rate; SV, stroke volume; CO, cardiac output; AIx, augmentation index; PWV, pulse wave velocity. Note: bars represent mean values, dots represent individual patients. * *p* = 0.034; ** *p* = 0.015; *** *p* = 0.013; **** *p* = 0.001.

**Figure 3 jcm-12-00029-f003:**
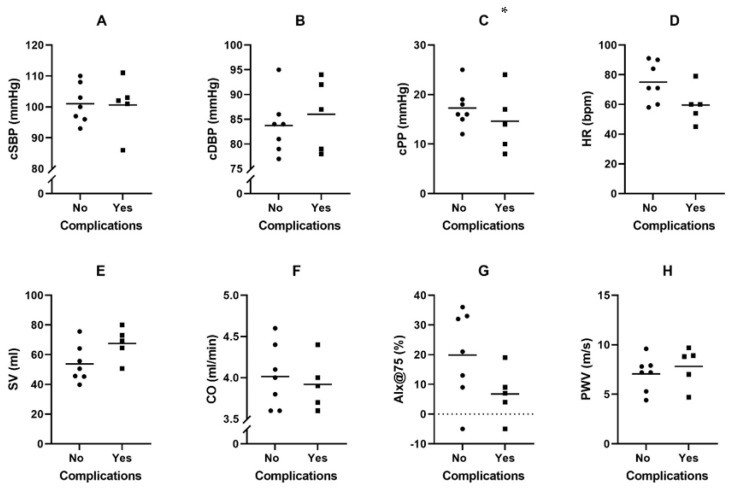
Distribution of non-invasive hemodynamics in HF with an LVAD with and without adverse events (**A**–**H**). Abbreviations: cSBP, central systolic blood pressure; cDBP, central diastolic blood pressure; cPP, central pulse pressure; HR, heart rate; SV, stroke volume; CO, cardiac output; AIx, augmentation index; PWV, pulse wave velocity. Note: bars represent mean values, dots represent individual patients. * *p* = 0.044.

**Table 1 jcm-12-00029-t001:** Baseline characteristics of HF with an LVAD with successful or failed measurement.

Characteristics	Successful Measurement (12)	Failed Measurement (35)	*p*-Value
Age, years, IQR	57 (34–66)	60 (53–64)	ns
Sex, men, *n* (%)	10 (83.3)	30 (85.7)	ns
Weight, kg	80 (78–96)	88.2 (77.6–101.2)	ns
BMI, kg/m^2^	26.6 (25.3–30)	27.8 (25.3–30.0]	ns
Time on LVAD, months	19.5 (3–30)	23 (13–41]	ns
Etiology of HF, ischemic, % (*n*)	58.3 (7)	57.1 (20)	ns
Smoking, %	2 (16.7)	2 (5.7)	ns
Diabetes, %	2 (16.7)	5 (14.3)	ns
Hypertension, %	2 (16.7)	6 (17.1)	ns
LVAD device type, HM-3, % (*n*)	11 (91.7)	35 (100)	ns
SBP, mmHg	108 (104–116)	102 (99–112)	ns
MAP, mmHg	93 (84–96)	85 (82–90)	ns
DBP, mmHg	83 (75–87)	79 (73–84)	ns
PP, mmHg	23 (17–28)	26 (23–30)	ns
LVAD parameters			
• RPM, r/min	5300 (5200–5400)	5400 (5300–5500)	ns
• Flow, L/min	4.4 (4.1–4.5)	4.6 (4.3–4.9)	0.014
• PI	4.4 (3.9–5.7)	3.6 (2.9–4.40)	0.002
• Pulse Power, Watt	3.8 (3.8–4)	4.0 (3.8–4.2)	ns
eGFR, ml/min/1.73 m^2^	69 (63–90)	60.0 (43–77)	ns
Hb, mmol/L	8.7 (7.7–9)	8.6 (7.8–9.3)	ns
Bilirubin, µmol/L	9.0 (8–16)	11.0 (8–17)	ns
NT-proBNP, pmol/L	152 (99–213)	190 (138–390)	0.057

Abbreviations: LVAD, left ventricular assist device; BMI, body mass index; HF, heart failure; HM-II, Heartmate II; HM-III, Heartmate III; SBP, systolic blood pressure; MAP, mean arterial pressure; DBP, diastolic blood pressure; PP, pulse pressure; RPM, revolutions per minute; PI, pulsatility index; eGFR, estimated glomerular filtration rate; Hb, hemoglobin; NT-proBNP, N-terminal pro-brain natriuretic peptide.

**Table 2 jcm-12-00029-t002:** Baseline characteristics of HF with an LVAD in different age categories.

Characteristics	Age < 55 Years (*n* = 4)	Age ≥ 55 Years (*n* = 6)	*p*-Value
Age, years, IQR	33 (24.5–49)	62 (57–72)	<0.001
Sex, men, *n* (%)	4 (100)	6 (75)	ns
Weight, kg	81.7 (77.9–112.6)	80 (78–96)	ns
BMI, kg/m^2^	25.7 (29.4–25.2)	27.9 (25.7–29.9)	ns
Time since LVAD, months	29 (14.3–66.3)	6.0 (2–25)	ns
Etiology of HF, ischemic, *n* (%)	0	7 (87.5)	0.004
Smoking, %	0	2 (25)	ns
Diabetes, %	1 (25)	1 (12.5)	ns
Hypertension, %	0	2 (25) 25% (2)	ns
LVAD device type, HM-III, % (*n*)	1 (25)	0	ns
SBP, mmHg	108.5 (98.3–117.3)	108 (104–116)	ns
MAP, mmHg	90 (78.8–96)	93 (87–95)	ns
DBP, mmHg	79.5 (69.8–88.5)	83 (80–87)	ns
PP, mmHg	25 (21.3–33.3)	23 (15–28)	ns
LVAD parameters			
• RPM, r/min	5250 (5200–7350)	5300 (5250–5250)	ns
• Flow, L/min	4.4 (4.1–4.5)	4.4 (4.2–4.5)	ns
• PI	5.1 (4.5–6.4)	4.1 (3.9–4.6)	ns
• Pulse Power, Watt	4.0 (3.8–4.8)	3.8 (3.8–4.0)	ns
eGFR, m/min/1.73 m^2^	75.7 (67.5–86)	63 (54–83.5)	ns
Hb, mmol/L	9.0 (8.6–9.3)	8.4 (7.7–8.9)	ns
Bilirubin, µmol/L	16 (7.5–29)	9 (8.5–10.5)	ns
NT-proBNP, pmol/L	117.5 (39.5–201)	152 (120–215)	ns

Abbreviations: LVAD, left ventricular assist device; BMI, body mass index; HF, heart failure; HM-II, Heartmate II; HM-III, Heartmate III; SBP, systolic blood pressure; MAP, mean arterial pressure; DBP, diastolic blood pressure; PP, pulse pressure; RPM, revolutions per minute; PI, pulsatility index; eGFR, estimated glomerular filtration rate; Hb, hemoglobin; NT-proBNP, N-terminal pro-brain natriuretic peptide.

**Table 3 jcm-12-00029-t003:** Baseline characteristics of patients with an LVAD with and without any adverse event.

Characteristics	LVAD without Adverse Event (7)	LVAD with Any Adverse Event (5)	*p*-Value
Age, years	57.5 (29.5–63)	55 (44–72)	ns
Sex, men, *n* (%)	5 (71.4)	5 (100)	ns
Weight, kg	78.5 (77.5–86.6)	96 (79.5–110.1)	ns
BMI, kg/m^2^	25.8 (24.7–28.1)	30 (25.9–30.4)	ns
Time since LVAD, months	13.5 (1.75–28)	28.3 (7.5–53)	ns
Etiology of HF, ischemic, % (*n*)	57.1 (4)	60 (3)	ns
Smoking, %	2 (28.6)	0	ns
Diabetes, %	0	2 (40)	ns
Hypertension, %	2 (28.6)	0	ns
LVAD device type, HM-III, % (*n*)	6 (100)	4 (80)	ns
SBP, mmHg	112 (102–118.3)	107 (102–113)	ns
MAP, mmHg	93.5 (84.5–97.8)	91 (777–95.5)	ns
DBP, mmHg	82.5 (77–89.3)	83 (65.5–88)	ns
PP, mmHg	20 (14.8–29.8)	24 (21.5–32.5)	ns
LVAD parameters			
• RPM, r/min	5300 (5175–5300)	5400 (5200–5400)	ns
• Flow, L/min	4.3 (3.9–4.4)	4.5 (4.3–4.5)	ns
• PI	4.2 (3.7–7.6)	4.4 (4.2–5.3)	ns
• Pulse Power, Watt	3.8 (3.7–4.0)	3.9 (3.8–4.7)	ns
eGFR, mL/min/1.73 m^2^	73 (58.5–84)	66 (54–90)	ns
Hb, mmol/L	8 (7.7–9.5)	9 (8.6–9)	ns
Bilirubin, µmol/L	9 (6.8–12.8)	12 (8–25)	ns
NT-proBNP, pmol/L	162 (90.2–256.5)	152 (66–212)	ns

Abbreviations: LVAD, left ventricular assist device; BMI, body mass index; HF, heart failure; HM-II, Heartmate II; HM-III, Heartmate III; SBP, systolic blood pressure; MAP, mean arterial pressure; DBP, diastolic blood pressure; PP, pulse pressure; RPM, revolutions per minute; PI, pulsatility index; eGFR, estimated glomerular filtration rate; Hb, hemoglobin; NT-proBNP, N-terminal pro-brain natriuretic peptide.

## Data Availability

Database of the Geriatric Medicine Division of the Erasmus MC University Medical Center, Rotterdam.

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
