# Peer review of "The Effect of Age on Non-Invasive Hemodynamics in Chronic Heart Failure Patients on Left-Ventricular Assist Device Support: A Pilot Study"

_jcm, 2022, doi:10.3390/jcm12010029_

Round 1

Reviewer 1 Report

this study is very interesting because focuses attention on the relationship between the hemodynamic and postoperative outcomes according to the age of the patients. Although the study design is well conducted the research has a lot of limitations. first of all very few patients entered the study. the comparison has been conducted between a very low number of groups so I can hardly understand how a statistical analysis might have been calculated, another limitation is the parameters analyzed. the grade of aortic regurgitation and mitral regurgitation have been missed. the right ventricular function is never mentioned. It is difficult to understand which is the sense to analyze the baseline characteristics of the patients excluded from the measurements. 

Author Response

We thank the reviewer for the comments, we acknowledge that the sample is relatively small, therefore we underline the exploratory aspect of this study. Further limitations of the study are stressed in the revised version of the MS (lines 188-190).

Also we agree with the fact that information on ventricular function is not given this was due to the fact that this was not one of the aims of the study.

We have shown the characteristics of the patients without successful measurements in order to illustrate completely, the characteristics of the population.

Reviewer 2 Report

The authors reports that LVAD patients aged 55 years and older had higher levels of aortic stiffness, augmentation index, and heartrate but had a lower stroke volume when compared with younger patients. The pulsatile component of blood pressure is decreased in patients with adverse events. 

First of all, the number is too small for scientific evaluation. Also the topic is not novel or significant. 

Author Response

We thank the reviewer for the comment, we acknowledge that the sample is relatively small, therefore we underline the exploratory aspect of this study.

Further limitations of the study are stressed in the revised version of the MS (lines 188-190).

Round 2

Reviewer 1 Report

The major revisions has not be conducted as required

Reviewer 2 Report

I think the updated version is good for publication.